# LADDER RESIDUAL: REDEFINING TENSOR PARALLELISM IN TRANSFORMERS FOR ACCELERATED INFERENCE

## ABSTRACT

Large language model inference is both memory-intensive and time-consuming, often requiring distributed algorithms to efficiently scale. Tensor parallelism (TP) is a common technique used in multi-gpu training and inference to partition computation across multiple devices, reducing memory load and computation time. However, such parallelism necessitates fast interconnects between the devices which has been a major bottleneck and limits the gains obtained by scaling up the number of devices. We introduce Ladder Residual, a simple architectural modification applicable to all residual-based models that enable straightforward overlapping that effectively hides the latency of communication. Our insight is that in addition to systems optimization, one can also redesign the model architecture to decouple communication from computation. For a Transformer model of 8B size, applying Ladder Residual to all its layers achieves 29% end-to-end wall clock speed up at inference time with TP world size of 8 devices. We refer to such model as the Ladder Transformer. We train a 1B and 3B Ladder Transformer from scratch and observe comparable performance to a standard dense transformer baseline. We also conduct adaptation experiments for our approach and show that it's possible to adapt parts of the Llama-3.1 8B model with minimal accuracy degradation by only retraining for 3B tokens. To further push the performance frontier, we propose another architectural modification which drops communications in the model, unlocking fast LLM inference in settings devoid of NVLink or other fast interconnects.

## 1 INTRODUCTION

With the rapid scaling of Large Language Models (LLMs) (Smith et al., 2022; Workshop et al., 2023; Brown, 2020), the compute and memory requirements for training and inference have grown significantly. Tensor parallelism (TP) (Shoeybi et al., 2020) is a widely used model parallelism technique that partitions the weights and intermediate activations across multiple GPUs. In contrast to pipeline parallelism (Narayanan et al., 2021) and data parallelism (Li et al., 2020), which rely on processing independent batches of user requests on each device, tensor-parallel inference enables multiple devices to cooperate to process a single batch of user requests at a time, therefore in theory allowing infinite scaling given a sufficient number of processors. However, TP requires synchronizing the partitioned intermediate activations across the GPUs. This synchronization is a blocking `AllReduce` operation on the activations across the GPUs and is therefore bottlenecked by the network communication latency. Even for GPUs connected via fast interconnects (like NVLink (NVIDIA Corporation, 2024)), the communication costs can account for 40% of the latency at inference time when running llama 3 8B with batch size 4 and TP world size of 8.

Past works have attempted to overlap the communication latency of TP by overlapping computation and communication. Chang et al. (2024) write fused kernels for `AllGather` followed by matmul and matmul followed by `ReduceScatter`. They break down matmuls into tiles and try to hide the latency of communicating a matmul tile with the computation of subsequent tiles. Jangda et al. (2022) propose CoCoNet, a domain-specific language to express distributed machine learning workloads. They propose to generate efficient GPU kernels for computation and communication using a custom compiler for the DSL. This approach has limited applicability with existing frameworks like PyTorch (Paszke et al., 2019) and JAX (Frostig et al., 2018) since the user needs to be well-acquainted with the DSL to generate efficient GPU kernels. Moreover, with the breakneck pace of accelerator and interconnect changes, these low-level systems optimizations require a rewrite for every new generation of hardware. However, there is a fundamental limit to how much communication latency can be reduced on such works which do not change the underlying model architecture. Instead of pure hardware optimizations (e.g., larger NVLink domain connecting 36 or 72 Blackwell GPUs) or pure low-level software optimization (e.g., rewriting all matmuls to overlap with communication), we explore model architectural changes that would enable

a reduction in communication latency while maintaining accuracy. This makes our approach quite simple to apply in practice using a high level machine learning framework like PyTorch (Paszke et al., 2019) or JAX (Frostig et al., 2018) without writing any low-level device code.

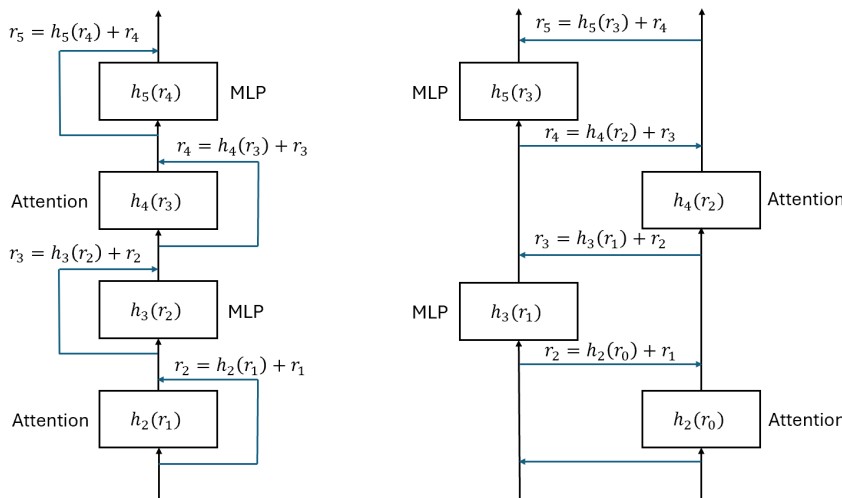

Figure 1: Illustration of a standard Transformer block (left) and a Ladder Residual block (right). The blue edge denotes the residual connection. In Ladder Residual, the residual connection remains the same while each module $h_i$ takes the stale input $r_{i-2}$.

Communication is blocking because there is a sequential structure between communication and computation in existing model designs: we wait for communication in order to prepare the correct input for the next computation. In the prevalent residual-based architectures, the computation flow can be written as $x_{i+1} = h_{i+1}(x_i) + x_i$, where $x_i$ is the residual stream after layer $i$ and $h_{i+1}$ is the computation at layer $i+1$. Notice that the communication of $x_i$ needs to be done before executing $h_{i+1}$ if $h_i$ is partitioned across devices. Liu et al. (2023b) found that activation changes slowly in Transformer, as the norm of each update $h_{i+1}(x_i)$ is small compared to the residual. Based on this observation, we hypothesize that maintaining the regular residual connection is enough to restrict the representation shift, and we can feed each module a "stale" input to create overlapping opportunities.

We propose *Ladder Residual*, a simple change where we reroute the residual stream after layer $i-1$ (instead of layer $i$) as input to layer $i+1$: $x_{i+1} = h_{i+1}(x_{i-1}) + x_i$. With this design, the computation of $h_{i+1}$ is decoupled from the communication of $x_i$, enabling straightforward overlapping to hide the latency of communication. Figure 1 shows how Ladder Residual can be applied to the Transformer architecture. At inference time with TP world size of 8 (i.e., across 8 devices), running a Transformer with Ladder Residual can be around 30% faster over the standard Transformer. In Table 1, we provide the inference speedup on Transformers of different sizes. The proposed Ladder Residual method is also fully compatible with Sequence Parallelism (SP) (Korthikanti et al., 2022) which helps in reducing the activation memory when training large models with long context lengths. Our method obtains 5-7% training speedup when training an 8B model with 8k context length on 64 H100s with 3D parallelism across the GPUs (Tensor Parallel, Sequence Parallel and Fully Sharded Data Parallel (FSDP) (Zhao et al., 2023; Rajbhandari et al., 2020)), but we decide to focus on the inference speed ups in this paper since training with just FSDP and SP and is usually faster because weights can be pre-fetched and gradient synchronization using `ReduceScatter` can be overlapped in FSDP making communications in FSDP non-blocking.

Because of the widespread use of Transformer (Vaswani, 2017) based Language models, we focus on applying Ladder Residual on the Transformers in this paper, and we call the resulting model as Ladder Transformer. We conduct experiments under two scenarios to verify if we can maintain the same performance after applying Ladder Residual to a Transformer-based model:

- **Pretraining from scratch**: We train a 1B and a 3B parameter Ladder Transformer model with 100B tokens on the FineWeb-edu dataset (Lozhkov et al., 2024) and compared it with the standard transformer of the same size trained on the same amount of tokens. We find that the Ladder Transformer matches the performance of the standard Transformer model.

- **Post-training adaptation**: We take the pretrained Llama-3.1-8B-instruct model (Dubey et al., 2024) and apply Ladder Residual on the upper half layers. We then train it on 3B tokens to adapt to the representation shift. With this relatively light retraining, we can obtain a hybrid Ladder Llama that is on par with the original Llama model on AlpacaEval, math, commonsense reasoning, and knowledge-intensive tasks while only being slightly worse on coding benchmarks.

In section 5, as a proof of concept of what Ladder Residual can unlock, we additionally propose *Desynced Residual*, an architectural modification that eliminates communication by restricting certain modules to act on the local activations on each device independently. We found that it's possible to eliminate 75% of the communication without much performance degradation when training a 1B model. Since communication can be too heavy to overlap in some scenarios (for example, with cross-node communication or even wireless communication), applying Ladder Residual can further unlock the potential of scaling TP to a giant GPU cluster.

Table 1: Inference speed up from applying Ladder Residual on Transformer. The test setup is 1024 prompt length, 256 generated tokens, batch size 1, and TP world size of 8. The speedup value is calculated by comparing Ladder Transformer with Standard Transformer's inference throughput in tokens per second. We measure the speedup with both with and without NVLink inteconnect.

| Model size | Ladder Transformer (No NVLink) | Ladder Transformer (NVLink) |
| --- | --- | --- |
| 1B | 1.51x | 1.38x |
| 3B | 1.38x | 1.24x |
| 8B | 1.56x | 1.29x |
| 70B | 1.37x | 1.17x |

## 2 BACKGROUND

We provide some background of Tensor Parallelism (TP), where the communication happens in TP, and how communication is a major latency bottleneck in standard Transformer.

### 2.1 TENSOR PARALLELISM

Tensor parallelism (Shoeybi et al., 2020) is a commonly used technique in distributed training/inference. It partitions weights and activations across devices and performs partial computations on each device. Consider a sequence of 2 linear layers with weight matrices $A$ and $B$ and input activation $X$ that is running on 2 GPUs (TP world size of 2), we split $A$ along the output dimension into $[A_1, A_2]$, and split $B$ along the input dimension into $\begin{bmatrix} B_1 \\ B_2 \end{bmatrix}$. Then the output of the sequence of the 2 linear layers can be computed as $(XA)B = (XA_1)B_1 + (XA_2)B_2$ and we effectively partition the computation on the two devices. The final summation requires an `AllReduce` operation to aggregate the partial sums on each device, which introduces communication overhead. The `AllReduce` overhead increases with increasing message size and increasing number of devices participating in the `AllReduce`. A transformer layer consists of an attention block and an MLP block: both can be considered as a sequence of two matrix multiplications and therefore fit into the tensor parallelism paradigm described above. Thus each transformer layer contains 2 `AllReduce` operations: one for attention and another for MLP. Denoting the input to the $i^{th}$ block as $x_{i-1}$, the transformer can be viewed as the following sequential structure:

$$\begin{aligned}
x_i^* &= h_i(x_{i-1}) \\
x_i &= \texttt{AllReduce}(x_i^*) + x_{i-1} \\
x_{i+1}^* &= h_{i+1}(x_i) \\
x_{i+1} &= \texttt{AllReduce}(x_{i+1}^*) + x_i
\end{aligned} \tag{1}$$

where the $*$ denotes a partial-sum that requires an `AllReduce` for the full output replicated across all the GPUs. Note that the `AllReduce` operation is the identity function for a model running on 1 GPU.

A Transformer with N layers needs to perform the `AllReduce` 2N times and this can account for 40% of the inference latency for a 8B model using TP world size of 8, even with NVLink interconnect. For communication without NVLink support or inter-node communication, their latency can account for around 85% of the end-to-end latency. Modern nodes contain GPUs connected via NVLink but usually have

no more than 8 GPUs per node, due to limited PCIe lanes or power and cooling constraints in datacenters. There is a steep falloff in communication bandwidth and latency when communication happens outside a node either over InfiniBand or Ethernet thus making scaling TP practically infeasible outside a node.

## 3 LADDER TRANSFORMER

In this section we propose our Ladder Residual approach applied to a Transformer model and validate that it can speed up inference.

---

**Algorithm 1** Ladder Transformer Layer with Tensor Parallelism. Note that the AsyncAllReduce returns a handle which is passed to the next layer.

---

```
 1: function LAYER(residual, prev_attn_out, prev_mlp_out,
                     prev_attn_handle, prev_mlp_handle)
 2:     prev_attn_handle.wait()
 3:     residual ← residual + prev_attn_out
 4:
 5:     attn_out ← AttentionNorm(residual)
 6:     attn_out ← Attention(attn_out)
 7:     attn_out, attn_handle ← AsyncAllReduce(attn_out)
 8:
 9:     prev_mlp_handle.wait()
10:     residual ← residual + prev_mlp_out
11:
12:     mlp_out ← MLPNorm(residual)
13:     mlp_out ← MLP(mlp_out)
14:     mlp_out, mlp_handle ← AsyncAllReduce(mlp_out)
15:
16:     return residual, attn_out, mlp_out, attn_handle, mlp_handle
17: end function
```

---

### 3.1 ARCHITECTURE DESCRIPTION

In Equation 1, the `AllReduce` operation is blocking the next block from execution since $h_{i+1}$ requires $x_i$ as the input. Ladder Residual mitigates this problem by routing the $x_{i-1}$ to block $h_{i+1}$, effectively making the input of $h_{i+1}$ independent of the output of the `AllReduce`, therefore allowing overlapping `AllReduce`$(x_i^*)$ with $h_{i+1}$.

Specifically, we change the computation flow of Equation 1 into:

$$
\begin{aligned}
x_i^* &= h_i(x_{i-2}) \\
x_i &= \texttt{AllReduce}(x_i^*) + x_{i-1} \\
x_{i+1}^* &= h_{i+1}(x_{i-1}) \\
x_{i+1} &= \texttt{AllReduce}(x_{i+1}^*) + x_i
\end{aligned}
\tag{2}
$$

Note that the residual stream of each block still takes the output from the previous module as usual, this ensures the module $i$ is still able to process output from all previous $i-2$ modules.

### 3.2 INFERENCE IMPLEMENTATION

**Ladder Residual Implementation**: We present the Ladder Transformer's layer's pseudo-code in Algorithm 1. To implement the Ladder Transformer, following the description of Equation 2 we call `AsyncAllReduce` for the `Attention`'s output. This returns a handle that can be used to synchronize the output to ensure that the `AsyncAllReduce` has finished. It should be noted that NCCL collectives in PyTorch always run on a different CUDA stream than the default compute stream used by PyTorch thus making them asynchronous. As soon as the `AsyncAllReduce` is called, we synchronize by calling wait on the previous layer's MLP's output and subsequently the CPU launches the kernels for `MLPNorm` and then `MLP` on the default compute stream and eventually calling the `AsyncAllReduce` for MLP. The handles for these NCCL operations are then passed onto the next layer which uses them for synchronization when needed.

**Alignment with Real-World Scenarios**: To evaluate the practical benefits of Ladder Residual, we integrated this mechanism into a standard Llama-like Transformer. Building upon `gpt-fast` (PyTorch

Labs, 2024), we partition the weights of the attention and feedforward modules for tensor parallelism to optimize inference speed. We use CUDA graphs (Coleman, 2020) to generate static computation graphs for both the prefill and decode phase to reduce CPU kernel launch overheads which can be a big bottleneck especially during the decode phase of Transformer inference. Additionally, we use FlashAttention (Dao, 2023) during prefill and FlashDecode (Dao et al., 2023) during the autoregressive decode phase of generation for accelerating inference.

### 3.3 FASTER INFERENCE WITH LADDER RESIDUAL

In this section, we benchmark Ladder Residual under various scenarios and show that across various model sizes, batch sizes, and varying TP world size, Ladder Residual can obtain considerable speedup over the standard Transformer. We also compare against the parallel attention and mlp design in  Chowdhery et al. (2022); Wang & Komatsuzaki (2021), which effectively cuts half of the communication. We also consider a scenario without NVLink, and show that our method can obtain more than 50% of the speedup when high-speed interconnect is not available.

#### 3.3.1 SETUP

We benchmark several algorithmic variants to evaluate their performance in large-scale language model inference. The candidates include:

- Standard Transformer: The standard transformer implementation.

- Parallel Attention and Mlp: Following the PaLM parallelization strategy Chowdhery et al. (2022); Wang & Komatsuzaki (2021), we fuse the weights of the query, key, value, gate, and up projections into a single matrix. The outputs are then split, and the attention and swiglu are performed in parallel. This effectively cut half of the communication, with the extra benefit of being able to fuse attention and mlp together. PaLM has shown its feasibility when trained from scratch, and we also experiment with it in subsection 4.1.

- Ladder Residual: The architectural optimization we propose to overlap computation with communication required for Tensor Parallelism.

- Communication-Free Upper Bound: An upper bound that removes all communication operations in the model to represent the theoretical maximum speedup achievable.

Note that different Transformer-based models usually have slightly different designs. The Transformer we used followed Llama-3's design and we designed the width and depth to meet specific parameter counts. The benchmarking results here won't be too different when moving to a different design since the communication patterns are mostly the same across various transformer variants.

To simulate realistic inference scenarios, we select multiple experimental configurations. Users typically set prompt lengths not exceeding 1024 tokens and generate short outputs upto 32 new tokens while using an LLM as a chat assistant. Therefore, we set the prompt length to 1024 and the generation length to 32 tokens.

We vary the tensor parallel world sizes among 1, 2, 4, and 8, and batch fsizes among 1, 4, 16, and 64 to understand performance under different generation settings. All models are benchmarked on 1, 2, 4 or 8 NVIDIA H100 GPUs.

To evaluate the impact of hardware communication capabilities, we adjust the NVLink settings using environment variables. We enable NVIDIA SHARP (Graham et al., 2016) by setting `NCCL_NVLS_ENABLE=1` and disable NVLINK communication by setting `NCCL_P2P_DISABLE=1`. This allows us to assess the performance of different algorithms in varying communication environments. All of results has been re-run for 3 times to ensure the results is stable and confident.

#### 3.3.2 BENCHMARKING

We characterize the inference efficiency improvements enabled by Ladder Residual in three different ways.

First, we measure the **best latency** achievable using both the Ladder Residual architecture and a traditional transformer baseline, both in terms of end-to-end latency and broken down by inference phase (*prefill* vs *decode*). The best latency in our setting is achieved using a batch size of 1 and a TP degree of 8 (the maximum TP world size possible on our 8-GPU node). We present the results of these latency-optimized experiments in Table 2. In this latency-optimized regime, both with and without NVLink, we find that the Ladder Residual architecture outperforms the baseline in both prefill and decode latency.

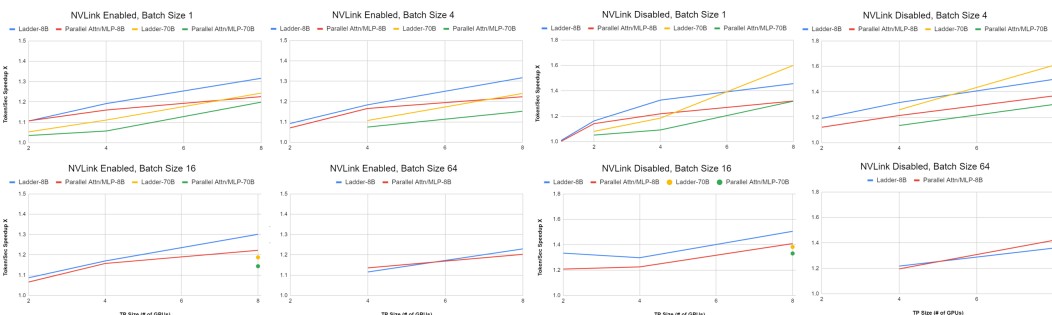

Figure 2: Improvement in end-to-end inference throughput achieved by communication-efficient architectures relative to a traditional transformer, benchmarked on Llama-3 8B and 70B. Running with NVLink, Ladder Residual architecture can achieve up to $24\%$ greater throughput than the traditional Transformer. Without NVLink, we observe speedups up to $60\%$. All experiments were conducted on a generation task with $1024$ prompt tokens and $512$ completion tokens. Some data points are missing due to OOM.

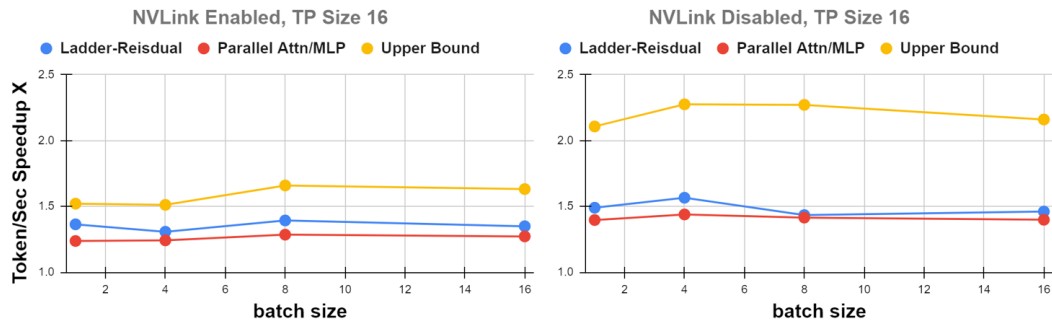

Figure 3: End-to-end inference throughput improvement on Llama-3-405B on a generation task with $1024$ prompt tokens and $512$ completion tokens. Here we use TP size 16 across two nodes each with 8 GPUs, connected with InfiniBand. "Upper Bound" denote the model without doing communication at all (which results in incorrect computation). Due to the high cost of cross-node communication, Ladder Residual architecture is able to achieve more than $30\%$ improvement across various batch size even with NVLink enabled adn around $50\%$ without NVLink.

Second, we measure the **throughput** achievable at each TP world size, and at a representative selection of batch sizes, for 8B and 70B size. We present the results of these experiments in Figure 2. In these throughput-oriented experiments, we again find that the Ladder Residual architecture significantly outperforms the traditional transformer architecture, and that these improvements are robust to the presence or absence of NVLink. We also find that the throughput gains from adopting the Ladder Residual architecture increase as the TP degree increases, reflecting the greater proportion of run time spent in communication relative to compute as we partition the computation across a larger number of devices. Lastly, we see the amount of improvement decreases as we increase from 8B to 70B when running with NVLink, as the computation scales faster than communication. However, the trend is reversed at large TP size when running without NVLink, likely due to the much higher cost of communication in that scenario.

Serving a model of larger size can be a challenge since even loading the model can require more than 8 GPUs. Cross-node TP communication is very expensive, however the common practice of using intra-node TP with cross-node Pipeline Parallelism (PP) is dependent on batch size to reduce gpu idle time (for example, with batch size = 1, half of the GPUs will be idle at any given time). With the speedup from Ladder Residual, cross-node TP can be a viable option. We benchmarked 405B under such setting in Figure 3, found that even for nodes with fast InfiniBand interconnect, Ladder Residual architecture can achieve more than $30\%$ throughput improvement across various batch sizes.

Finally, recognizing that different practitioners may wish to pursue different tradeoffs between latency and throughput depending on their application, we characterize the **latency/throughput Pareto frontier** achieved by each architecture in Figure 4. Consistent with typical deployment practice, we find that throughput per device for our 8B model is maximized when the model is served from just a single device,

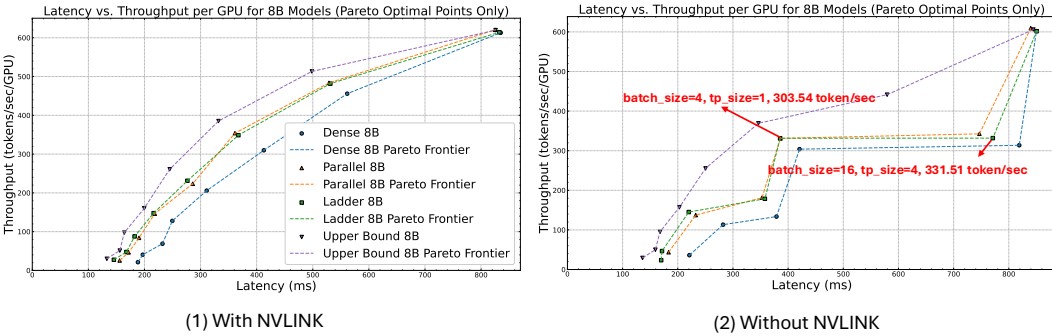

(1) With NVLINK            (2) Without NVLINK

Figure 4: Pareto frontier of completion latency vs aggregate throughput per GPU for different 8B-scale model architectures in a batched inference setting. For each architecture, we sweep over both batch size and TP world size to find the Pareto-optimal configurations. Because an 8B model fits on a single device, the configurations which maximize throughput are always those which use no TP (points on the top-right corner), in which case the differences in communication overhead between architectures disappear. However, when jointly optimizing for both latency and throughput, we find that TP is beneficial (all points before top-right), in which case the Ladder architecture achieves Pareto improvements over the standard transformer architecture. All experiments measure end-to-end time on a generation task with 1024 prompt tokens and 32 completion tokens per sequence.

in which case our communication efficiency interventions provide no benefit. However, when jointly optimizing throughput and latency, we find that adopting the Ladder Residual architecture significantly shifts the Pareto frontier towards more favorable latency/throughput tradeoffs compared to the standard transformer architecture. We note that that this demonstrates that even for a relatively small model of only 8B parameters, it can still be worthwhile to optimize the design of one's model to support efficient inference with high degrees of tensor parallelism.

Table 2: Detailed breakdown for prefill, decode and token/sec improvement (%) for different model. The speedup (%) calculated by using latency of optimized model divided by original model. All the experiment are based on batch size 1, TP world size of 8 GPUs.

| Model | Prefill Improvement (%) | Decode Improvement (%) | Token/sec Improvement (%) |
|---|---|---|---|
| NVLINK-UpperBound-Llama-8B | 37.00 | 28.69 | 41.66 |
| NVLINK-Parallel-Llama-8B | 16.93 | 17.30 | 21.00 |
| NVLINK-Ladder-Llama-8B | 12.46 | 23.71 | 29.65 |
| NO-NVLINK-UpperBound-Llama-8B | 50.72 | 43.44 | 78.72 |
| NO-NVLINK-Parallel-Llama-8B | 19.61 | 24.01 | 30.86 |
| NO-NVLINK-Ladder-Llama-8B | 24.40 | 30.71 | 43.01 |

## 4 Experiments and Results

We empirically verify our assumption that applying Ladder Residual does not hurt the performance. We show that Ladder Residual can be either used when training from scratch, or be applied to a pre-trained model with hybrid adaptation, and in both cases, the performance is on par with the original architecture.

### 4.1 Training from Scratch

We train a 1B and 3B Ladder Transformer from scratch and compare its performance with an equally sized standard Transformer model. All our models are trained on 100B tokens of FineWeb-edu dataset (HuggingFaceFW, 2024) using the StarCoder tokenizer (Li et al., 2023b). We also compare our model with the Parallel Attention/MLP architecture (Chowdhery et al., 2022; Wang & Komatsuzaki, 2021) which parallelizes the computation of the attention and the MLP module. This effectively reduces the communication cost by 50% for the tensor parallel all-reduce in both the forward and backward computation.

### 4.1.1 Experimental details

We use DDP (Distributed Data Parallel) (Li et al., 2020) to train the 1B and HSDP (Hybrid Sharded Data Parallel) (Zhao et al., 2023; Rajbhandari et al., 2020) to train the 3B models. For HSDP, we shard the model

within 1 node (equipped with 8x H100 GPUs) and replicate the model outside the node. We use mixed precision training (Micikevicius et al., 2018) in BF16 (Kalamkar et al., 2019) with gradient accumulation and gradient all-reduce/reduce-scatter in FP32 for training stability. We train all our models with 2048 context length with a batch size of 4M tokens in a batch. The models are trained with cosine scheduler with a warmup of 8B tokens to a peak learning rate of $3 \times 10^{-4}$. The learning rate is then decayed over 92B tokens to $3 \times 10^{-5}$.

We use EleutherAI's LM eval harness (Gao et al., 2024) to evaluate all our models on ARC (Clark et al., 2018), HellaSwag (Zellers et al., 2019), PIQA (Bisk et al., 2020), SciQ (Welbl et al., 2017) and Winogrande (Trinh & Le, 2018). We also evaluate Wikitext (Merity et al., 2017) perplexity for all our models.

### 4.1.2 RESULTS

The full results can be found at Table 3. We find that at the 1B model scale, the Ladder Transformer achieves obtains similar performance compared with the Standard Transformer while beating the Parallel Transformer. At the 3B parameter scale, we find that the Standard Transformer baseline model is better than the Ladder Transformer model with 3.2% lower perplexity and 1.2 points of absolute difference in accuracy. The Parallel Transformer has almost the same performance as the Ladder Transformer at the 3B scale.

Table 3: Performance of three architectures under two sizes, trained on FineWeb-edu for 100B tokens.

| Model | ARC-C | ARC-E | HellaSwag | PIQA | SciQ | Winogrande | Average | Wikitext PPL |
|---|---|---|---|---|---|---|---|---|
| Standard-Transformer-1B | 34.22 | 70.33 | 41.10 | 71.49 | 87.30 | 55.41 | 59.98 | 18.54 |
| Parallel-Transformer-1B | 30.46 | 67.97 | 40.35 | 71.16 | 87.40 | 55.17 | 58.75 | 18.95 |
| Ladder-Transformer-1B | 31.31 | 67.76 | 41.18 | 71.49 | 86.60 | 55.17 | 58.92 | 18.42 |
| Standard-Transformer-3B | 38.99 | 74.12 | 46.48 | 74.59 | 92.00 | 58.48 | 64.11 | 14.48 |
| Parallel-Transformer-3B | 38.48 | 73.02 | 45.55 | 73.67 | 90.00 | 57.46 | 63.03 | 14.96 |
| Ladder-Transformer-3B | 36.77 | 72.43 | 45.66 | 73.72 | 89.90 | 58.96 | 62.91 | 14.90 |

### 4.2 POST-TRAINING ADAPTATION

We investigate the feasibility of directly applying Ladder Residual on an existing pre-trained model, and we choose Llama-3.1 8B Instruct as the target. We applied Ladder Residual to the upper half of the Llama to keep the performance since we hypothesize that touching the lower layers can destroy knowledge that is hard to recover without large-scale retraining. We evaluate the adapted models on 8 benchmarks across a range of domains: accuracy on MMLU (5-shots) (Hendrycks et al., 2021) and ARC-Challenge (ARC-C, 25-shots) (Clark et al., 2018), normalized accuracy on OpenBookQA (OBQA) (Mihaylov et al., 2018), HellaSwag (HS, 10-shots) (Zellers et al., 2019), and TruthfulQA (TQ, mc1) (Lin et al., 2022). exact-match accuracy on GSM8K(GSM, 8-shots) (Cobbe et al., 2021), pass@1 on HumanEval+(HE+) (Chen et al., 2021), aggregated accuracy on IFEval (Zhou et al., 2023), and length controlled win rate (Dubois et al., 2024) against gpt4-turbo on AlpacaEval (Li et al., 2023c). The evaluation of HumanEval+ is conducted with EvalPlus (Liu et al., 2023a), AlpacaEval is done with the AlpacaEval2 library, and the rest of the evaluations are conducted with the LM-Evaluation-Harness library (Gao et al., 2024).

### 4.2.1 EXPERIMENTAL DETAILS

We convert a state-of-the-art open-source model, Llama-3.1-8B-Instruct into a hybrid Ladder Residual structure, by applying Ladder Residual to the upper half of the model (layers 16-32 for LLaMA-3.1-8B-Instruct). We call this variant Hybrid-Ladder-8B-16L in Table 4. We also experiment with more aggressive adaptation where we applied Ladder Residual to the layers 12-32 and we call this experiment Hybrid-Ladder-8B-20L. We conduct supervised fine-tuning (SFT) for the resulting model on the 7M subset and the Gen subset of the Infinity-Instruct dataset[1], which contains 3B tokens. We train for 2 epochs with AdamW optimizer with a batch size of 32. We use 5e-6 learning rate with 200 steps of linear warmup, followed by cosine annealing to the end. We use Axolotl [2] for our SFT experiments.

As shown in Table 4, after adaptation, there is a huge performance drop mainly on generative tasks as the computation flow is messed up. But after light retraining, the hybrid Ladder Llama is able to reach the same level of performance with the original Llama. By applying Ladder Residual on the last 16 layers (half of the 32 layers), we are able to obtain 14.5% end-to-end wall clock speed up at the inference time with TP world size of 8 and batch size of 1. Our results demonstrate the potential of Ladder Residual being

---

[1] https://huggingface.co/datasets/BAAI/Infinity-Instruct
[2] https://github.com/axolotl-ai-cloud/axolotl

Table 4: All models are either LLama-3.1 models or are adapted from Llama-3.1 8B Instruct in this table. Performance comparison across various benchmarks. Zeroshot denotes directly applying Ladder Residual without any retraining. **xL** denotes how many layers of the Llama-3.1-8B-Instruct are applied Ladder Residual.

| Model | MMLU | ARC-C | OBQA | HS | TQ | GSM | HE+ | IE | AE | Average |
|---|---|---|---|---|---|---|---|---|---|---|
| Llama-3.1-8B-Instruct | 68.14 | 60.32 | 43.00 | 80.04 | 36.84 | 84.99 | 60.40 | 52.57 | 18.69 | 56.11 |
| Hybrid-Ladder-8B-16L-zeroshot | 63.19 | 56.57 | 42.60 | 77.70 | 35.50 | 10.54 | 30.50 | 46.25 | 11.99 | 41.65 |
| Hybrid-Ladder-8B-16L-retrained | 65.93 | 59.13 | 42.20 | 78.86 | 39.66 | 80.29 | 59.10 | 59.02 | 21.95 | 56.24 |
| Hybrid-Ladder-8B-16L-distill | 67.49 | 60.24 | 44.20 | 79.03 | N/A | 82.56 | N/A | N/A | N/A | N/A |
| Hybrid-Ladder-8B-20L-retrained | 62.31 | 59.90 | 42.60 | 77.49 | 36.72 | 76.19 | 48.80 | 59.05 | 21.72 | 53.86 |

a drop-in adaptation technique to make the model faster without sacrificing performance. We additionally experiment with applying Ladder Residual to the last 20 layers of Llama and found that it leads to a slight drop in performance. There is a chance that with longer adaptation, or smarter adaptation techniques like distillation or iterative training, we can obtain a Ladder-Llama with more layers adapted. We leave the further exploration to future work.

## 5    ELIMINATING COMMUNICATION WITH DESYNCED RESIDUAL

We find that in settings where there is no NVLink connection, the latency for `AllReduce` dominates and its not possible to hide the latency of the `AllReduce` completely with the computation of another block in the Ladder Residual method. We thus propose an alternative method, Desynced Reisudal, that entirely drops the `AllReduce` communication and lets each device process its own activations independently. We call this architecture Desynced Residual because dropping the communication leads to desynchonization of the residual stream in the model which is re-synchronized at the next `AllReduce` operation. We experiment with 2 different Desync configurations: Desync Residual-2x and Desync Residual-4x. Desync Residual-$nx$ means that we only retain the last `AllReduce` operation in a group of $n$ sequential `AllReduce` operations i.e $(n-1)$ `AllReduce` operations are dropped from the model. While its possible to use an arbitrary communication pattern, we find that dropping `AllReduce` for Attention yields a model with lower Wikitext perplexity than dropping `AllReduce` for MLP while offering a good tradeoff between inference speed and model accuracy. We compare the latency improvement of Desyncd-nx and Ladder Residual in Table 6. While Desync-Residaul-2x performs slightly worse than Ladder Residaul, Desynced-Residaul-4x is able to achieve a larger gain, especially under the no NVLink setup (39% for Desynced-Residual-4x compare with 23% for Ladder Residual). The promising speedup shows the potential of Desync Residual as an alternative architecture to be further explored espeically in cases where communictaion can be very expensive.

### 5.1    EXPERIMENTS AND RESULTS

We run both 1B and 3B pretraining from scratch experiments for Desync Residual-2x and Desync Residual-4x models in the same setting as described in subsection 4.1. We find that the Desync Residual-4x model performs better than Desync Residual-2x model on 1B scale on Wikitext perplexity while being slightly worse on accuracy. While on the 3B scale, the Desync Residual-4x model is better on both perplexity and average accuracy. This is quite surprising and demonstrates that its possible to train models using the Desync Residual in a way to significantly reduce communication. For instance, the Desync Residual-4x drops 75% communications with almost no drop in model performance.

In our benchmarking results, we find that at a large batch sizes (64 in our experiments) with a TP world size of 8, it is possible to achieve around 30% improvement in inference throughput over NVLink and around 40% improvement without NVLink using the Desync Residual-4x architecture. We also observe a significant improvement in the first token latency (prefill latency) using the Desync Residual-4x model over Ladder Residual and Desync Residual-2x. The first token latency is especially important in scenarios where the LLM might be used as a classifier for example or in cases where multiple calls to the LLM need to be issues (agentic workflows for example). By manipulating where to communicate, Desync Residual allows designing networks that can drop arbitrary percent of the communication, which allows flexible architecture design giving different needs. For setup with very slow communication (for example wireless communication), Desync Residual enables sacrificing a small amount of performance in exchange for speed.

## 6    RELATED WORK

**Communication overlapping in parallelism**    Overlapping communication has been a widely explored area in prior works in order to achieve higher performance for distributed training. For Tensor Parallelism,

Table 5: Performance of Desync Residual compared to the Standard Transformer model trained on FineWeb-edu for 100B tokens.

| Model | ARC-C | ARC-E | HellaSwag | PIQA | SciQ | Winogrande | Average | Wikitext PPL |
|---|---|---|---|---|---|---|---|---|
| Standard-Transformer-1B | 34.22 | 70.33 | 41.10 | 71.49 | 87.30 | 55.41 | 59.98 | 18.54 |
| Desync Residual-2x-1B | 32.51 | 69.36 | 40.53 | 72.03 | 86.00 | 56.04 | 59.41 | 18.70 |
| Desync Residual-4x-1B | 32.17 | 68.60 | 41.25 | 71.60 | 87.40 | 54.70 | 59.29 | 18.58 |
| Standard-Transformer-3B | 38.99 | 74.12 | 46.48 | 74.59 | 92.00 | 58.48 | 64.11 | 14.48 |
| Desync Residual-2x-3B | 37.97 | 73.57 | 46.56 | 74.43 | 90.30 | 58.80 | 63.61 | 14.67 |
| Desync Residual-4x-3B | 38.57 | 73.15 | 46.49 | 73.72 | 91.90 | 58.64 | 63.75 | 14.60 |

Table 6: Detailed breakdown for prefill, decode and token/sec improvement (%) for different model. The speedup (%) calculated by using latency of optimized model divided by original model. All the experiment are based on batch size 64, TP degree 8.

| Model | Prefill Improvement (%) | Decode Improvement (%) | Token/sec Improvement (%) |
|---|---|---|---|
| NVLINK-UpperBound-Llama-8B | 25.43 | 27.22 | 34.31 |
| NVLINK-Ladder-Llama-8B | 9.93 | 18.70 | 14.43 |
| NVLINK-Desync-Residual-2x-Llama-8B | 8.40 | 12.03 | 10.51 |
| NVLINK-Desync-Residual-4x-Llama-8B | 17.07 | 16.53 | 20.36 |
| NO-NVLINK-UpperBound-Llama-8B | 31.60 | 51.95 | 64.97 |
| NO-NVLINK-Ladder-Llama-8B | 15.07 | 26.26 | 23.98 |
| NO-NVLINK-Desync-Residual-2x-Llama-8B | 13.36 | 24.36 | 21.59 |
| NO-NVLINK-Desync-Residual-4x-Llama-8B | 22.11 | 37.73 | 39.01 |

prior works (Jangda et al., 2022; Wang et al., 2022; NVIDIA, 2023) decompose the communication into more fine-grained operations in order to find computations with no dependency to overlap. Our work doesn't rely on such decompositions and therefore doesn't require Sequence Parallelism to handle the partitioned activations before all-gather. In FSDP (FairScale authors, 2021), the all-gather communication is usually prefetched to be overlapped with the communication. Pipeline Parallelism (NVIDIA, 2023; Li et al., 2023a; Lamy-Poirier, 2023) on the other hand, chunks the data into mini-batches which creates more opportunity for overlapping. Compared to these other parallelism approaches, TP has the advantage to be independent from the batch size or sequence length, and is able to partition the computation as much as possible given enough GPUs in theory.

**Efficiency-aware architecture improvements**     Prior works have explored various alternative designs for Transformer, for example parallel attention and mlp (Chowdhery et al., 2022; Wang & Komatsuzaki, 2021), linear attention (Katharopoulos et al., 2020), Grouped Query Attention (Ainslie et al., 2023), Cross-Layer Attention (Brandon et al., 2024) to improve the training and inference efficiency. Some of these variants are more widely adopted than others, due to the degree of impact they have on performance and efficiency. Past works have also considered adapting an existing checkpoint to these efficient variants. Ainslie et al. (2023) extracted grouped-query attention from a multi-head attention model, and Wang et al. (2024) converted a Llama model to a Mamba model. It retrained on 50B tokens to close the performance gap where our adaptation is much lighter (3B tokens), showing that the representation shift introduced by Ladder Residual is easier to recover. Wang et al. (2024) considered converting a Llama model to a Mamba model and used distillation to retrain the converted model. Such training paradigms that specifically tunes the model to align the original model could further improve the Ladder Residual based models.

## 7    CONCLUSION

We introduce Ladder Residual and Desync Residual, architectural modifications that allow overlapping communication with computation or dropping it entirely when running Tensor Parallelism. When applying Ladder Residual to Llama-3.1 8B Instruct, we only need lightweight retraining to reach the same level of performance with the original model, showing its potential to be a plug-in for any pretrained Transformer and we are able to obtain 14.5% end-to-end wall clock speed up at the inference time with TP world size of 8 and batch size of 1. We also trained a 1B and 3B Ladder Transformer from scratch, and find that they are comparable to the standard Transformer of the same size while achieving 25% speedup. On the other hand, Desync Residual provides flexible removal of communication when designing an architecture to train from scratch, which can be helpful in the case where the communication is too costly to overlap. Given that such a simple architectural change can obviate the need for expensive interconnects while maintaining model quality, we hope that our methods will inspire even closer co-design between model architecture and inference systems.

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
