# OpenReview forum: "Ladder Residual: Redefining Tensor Parallelism in Transformers for Accelerated Inference"
_ICLR.cc/2025/Conference — Submitted to ICLR 2025_

### Official Review · Reviewer_Jbio · 2024-10-16

**Soundness:** 4
**Presentation:** 1
**Contribution:** 2
**Rating:** 3
**Confidence:** 5

**Summary:**

This paper presents Ladder Residual, an algorithmic improvement on the traditional Transformer architecture. Ladder Residual delays each residual connection one MLP/Attention module later. This makes the communication introduced by Tensor Parallel no longer on the critical path, and can be overlapped with the computation.
The authors evaluated this method on two ways: 1. pretraining a 1B- and a 3B-parameter Ladder Residual Transformers from scratch with 100B tokens; 2. Adapting the architectural modification on a state-of-the-art Transformer model and finetune with only 3B tokens to adapt the shift. Models pretrained/finetuned from both methods show a promising performance, outperforming the parallel attn/mlp on both throughput and quality.
The authors further introduced fully desynchronized Residual, showing that it can also reach a satisfying performance with more throughput improvement in the case without NVLink.

**Strengths:**

- The authors introduce a clean method that can significant improve the communication-computation overlap opportunity and make the execution time shorter, while the performance shows a trivial drop.
- The authors perform a thorough evaluation on both model inference throughput/latency/pareto and the performance.
- The paper is easy to follow and clear to understand. The authors first introduced the background of tensor parallel and its communication cost, then elaborated their variant Ladder Transformer and explained why it solves the problem. Then they provides experiments on both inference throughput and model quality to prove the speedup and the performance under this method.

**Weaknesses:**

- Although Ladder Transformer is friendly to Tensor Parallel, it is not friendly to Pipeline Parallel, which is also necessary during the training process and the inference of large models, because both `prev_attn_out` and `prev_mlp_out` should be sent from a pipeline stage to the other. This prevents the Ladder Transformer from scaling up to a larger size. It would make the contribution more solid if the authors can discuss how to address this concern or why it may not be critical.
- The experiment details for Inference throughput/latency are not clear to me. Answering the following questions in the paper could make the experiment section easier to follow:
  - When NVLink is disabled, is NVLS still turned on?
  - What is the precision used during the inference?
  - What is the memory of a single H100 GPU? For Figure 2, when `Batch Size = 64`, why there is no number reported for TP=1 and 2?
    - I noted that the memory consumption in this case can be approximated as `8B(parameter) + 64(requests) * 1024(seq length) * 8(key/value heads) * 128(head hidden size) * 2(key and value) * 32(layers)=12B` numbers. Following the custom of using FP16 in model inference, this only consumes 24GB of memory, which leaves enough space for intermediate activations even on a single H100 GPU.
- The model size used in Figure 2 (8 billion parameters) is very small, compared to the capacity of hardware (up to 8xH100 GPUs). In this case, running the decode operation with a small batch size makes the computation very sparse, and thus the evaluation setup is not representative (as a [reference](https://lmsys.org/blog/2024-07-25-sglang-llama3/), even using A100 GPU can reach the regime of 2000 tokens/s/GPU, while the highest throughput in this paper is in Figure 3.(1), slightly above 600 tokens/s/GPU). It would make the result more convincing to measure the latency/throughput of a larger model (e.g. 65B) or batch size (e.g. 256 or 512). Besides, it would be beneficial to also report the baseline absolute number, since all other latency numbers are reported in the form of a relative improvement.
- For results in Table 3, it seems like Parallel Transformer is catching up with Ladder Transformer (Average from -0.17 to +0.12, Wikitext PPL from -0.53 to -0.06), as the number of parameters scales up from 1B to 3B, this raises concerns on the scalability of Ladder Transformer. Providing additional experiments with a larger model size or an explanation for why this trend might not continue could be helpful in supporting the significance of the Ladder Transformer.

**Questions:**

- In Figure 2, for the case Without NVLink, how to explain the reason of a throughput drop of batch size 4 and 16, when the TP world size shifts from 2 to 4?
- How to understand the correlation of the 3 columns in Table 2?
  - My understanding is that token/sec = 1(batch size) * 32(generation length, according to 3.3.1) / latency = 32 / (prompt + decode).
  - In this way, if the prompt improves x% and decode improves y%, the token/sec improvement is at most `1 / (1 - max(x%, y%)) - 1`, which equals `y% / (1-y%)` for all rows in this table because decode improvement is always higher. For most rows in this table, this approximation is very similar to the actual value (with less than 4% error), meaning that the decode time dominates the overall latency. However, in the first row it shows a 17% error.

Below is not relevant to my score of this paper, but only for paper readability:
- The Parallel Attn/MLP, as a commonly used baseline, is not well described. Adding more explanation/figure can improve the paper's self-completeness.
- Section 4 has too many grammar mistakes and is thus hard to follow.
- It would be better to have a breakup of computation (both MLP and Attention) and communication (both with and without NVL) to help understand the importance of the overlapping, as well as how much overlapped can be, and is achieved with Ladder Transformer.

---

> ### Author Response · Authors · 2024-11-21
> **Benchmarking speedup under faster framework, and results for 70B and 405B**
>
> > (as a reference, even using A100 GPU can reach the regime of 2000 tokens/s/GPU, while the highest throughput in this paper is in Figure 3.(1), slightly above 600 tokens/s/GPU). Besides, it would be beneficial to also report the baseline absolute number, since all other latency numbers are reported in the form of a relative improvement.
>
> We previously did all the benchmarks under CudaGraph + flash attention, since with torch compile the communication and computation will not be launched into different streams. Recently we resolved this issue with torch compile new features (which were released several days ago - by setting ```torch._inductor.config.reorder_for_compute_comm_overlap = True)``` and were able to benchmark under torch compile. With batch size 64, we are able to obtain 2463 token/sec on one H100 80G GPU. We redo all benchmarking under torch compile and found that **while the overall trend is the same, the speedup relative to the standard Transformer is even larger for all methods due to increased latency portion from the communication**. The result for 8B can be found below:
>
> Note that in all experiments below, we switch to 512 decoded tokens to make better token/sec comparison with https://lmsys.org/blog/2024-07-25-sglang-llama3/.
>
> ---
>
> NVLink=True
> | | bs=1,tp=4 | bs=1,tp=8 | bs=4,tp=1 | bs=4,tp=8 | bs=16,tp=4 | bs=16,tp=8 | bs=64,tp=4 | bs=64,tp=8 |
> | ----------- | ----------- | ----------- | ----------- |  ----------- |  ----------- |  ----------- |  ----------- |  ----------- |
> | Ladder Residual | 1.192x | 1.316x | 1.184x | 1.317x | 1.170x | 1.301x | 1.116x | 1.229x |
> | Ladder Residual, compile | 1.248x | 1.420x | 1.241x | 1.442x | 1.182x | 1.370x | 1.091x | 1.232x |
> | Parallel attn/mlp | 1.160x | 1.226x | 1.166x | 1.224x | 1.157x | 1.222x | 1.136x | 1.202x |
> | Parallel attn/mlp, compile | 1.233x | 1.333x | 1.207x | 1.317x | 1.175x | 1.296x | 1.120x | 1.210x |
>
> See https://anonymous.4open.science/r/ICLR2025_rebuttal-B81D/README.md for results under more batch size and tp size, as well as results under NVLink=False
>
> ---
>
> These findings show that despite we were previously using a slower benchmarking setting, as the overall trend doesn’t change when we switch to a faster setting, our conclusion in the paper still holds.
>
> > The model size used in Figure 2 (8 billion parameters) is very small, compared to the capacity of hardware (up to 8xH100 GPUs). In this case, running the decode operation with a small batch size makes the computation very sparse, and thus the evaluation setup is not representative. …(split into another response below)... It would make the result more convincing to measure the latency/throughput of a larger model (e.g. 65B) or batch size (e.g. 256 or 512)
>
> We did benchmark 70B size in table 1 and we agree with the reviewer that tensor parallelism is more interesting to study on larger models. To provide more comprehensive results, we benchmarked 70B, 405B with various batch sizes and TP sizes below:
>
> ---
>
> 70B results, NVLink=True:
> | | bs=1, tp=4 | bs=1, tp=8 | bs=4, tp=8 | bs=16, tp=8 |
> | ----------- | ----------- | ----------- | ----------- |  ----------- |
> | Ladder Residual | 1.111x | 1.244x | 1.240x | 1.188x |
> | Parallel attn/mlp | 1.057x | 1.120x | 1.152x | 1.144x |
>
> ---
>
> For 405B, even loading the model in bf16 requires > 800GB GPU memory, therefore we only benchmark under TP=16 (2 nodes, each with 8 H100 GPUs) for various batch sizes. Below are results with NVLink=True:
>
> | | bs=1 | bs=4 | bs=8 | bs=16 |
> | ----------- | ----------- | ----------- | ----------- |  ----------- |
> | Ladder Residual | 1.364x | 1.308x | 1.393x | 1.349x |
> | Parallel attn/mlp | 1.238x | 1.242x | 1.286x | 1.272x |
>
> See https://anonymous.4open.science/r/ICLR2025_rebuttal-B81D/README.md for results under more batch size, tp size, as well as results without NVLink.
>
> ---
>
>
> **Our method (Ladder Residual) still achieves significant speedup for both 70B and 405B and consistently outperforms the parallel attn-mlp baseline on both speed and model quality (shown on smaller size in the paper due to computation constraints)**. Note that the relative improvement of 70B is smaller than 8B size, since the computation scales faster than communication as model size increases (bits to be communicated scale linearly, while computation can scale quadratically). However, when we use TP=16, where the communication needs to happen across nodes, the improvement is larger as communication is more expensive. We expect these heterogeneous networking settings to become prevalent in the future as models get larger and inference hardware become more diverse.

---

> ### Author Response · Authors · 2024-11-21
> **Compatibility with Pipeline Parallelism**
>
> > Although Ladder Transformer is friendly to Tensor Parallel, it is not friendly to Pipeline Parallel, which is also necessary during the training process and the inference of large models, because both prev_attn_out and prev_mlp_out should be sent from a pipeline stage to the other. This prevents the Ladder Transformer from scaling up to a larger size. It would make the contribution more solid if the authors can discuss how to address this concern or why it may not be critical.
>
> It is possible to use Pipeline Parallel (PP) with the Ladder architecture. There are 2 ways in which PP can be used for Ladder Transformer model:
> Just before the pipeline boundary, we wait for the async AllReduces to complete, and send 3 tensors to the next pipeline stage: residual tensor, current_mlp_out tensor and current_attention_out tensor. It should be noted that this is still pretty cheap since generally the P2P communication is latency bound and can be implemented easily using the batch_isend_irecv API (https://pytorch.org/docs/stable/distributed.html#torch.distributed.batch_isend_irecv).
>
> No PP:
> ```
> def forward(
>    self,
>    previous_attention_out: Tensor,
>    previous_mlp_out: Tensor,
>    residual: Tensor,
>    attention_handle,
>    mlp_handle,
> ) -> Tensor:
>    attention_handle.wait()
>    residual = residual + previous_attention_out
>
>
>    current_attention_out = self.attention(self.attention_norm(residual))
>    current_attention_out = all_reduce(current_attention_out, async_op=True)
>
>
>    mlp_handle.wait()
>    residual = residual + previous_mlp_out
>
>
>    current_mlp_out = self.feed_forward(self.ffn_norm(residual))
>    current_mlp_out, mlp_handle = all_reduce(current_mlp_out, async_op=True)
>
>
>    return current_attention_out, current_mlp_out, residual, attention_handle, mlp_handle
> ```
> PP:
> ```
> def forward(
>    self,
>    previous_attention_out: Tensor,
>    previous_mlp_out: Tensor,
>    residual: Tensor,
>    attention_handle,
>    mlp_handle,
> ) -> Tensor:
>    attention_handle.wait()
>    residual = residual + previous_attention_out
>
>
>    current_attention_out = self.attention(self.attention_norm(residual))
>    current_attention_out = all_reduce(current_attention_out, async_op=True)
>
>
>    mlp_handle.wait()
>    residual = residual + previous_mlp_out
>
>
>    current_mlp_out = self.feed_forward(self.ffn_norm(residual))
>    current_mlp_out, mlp_handle = all_reduce(current_mlp_out, async_op=True)
>
>
>    if is_last_layer_on_pp_stage:
>        attention_handle.wait()
>        mlp_handle.wait()
>
>
>    return current_attention_out, current_mlp_out, residual, attention_handle, mlp_handle
> ```

---

> ### Author Response · Authors · 2024-11-21
> **Clarifications 1**
>
> > When NVLink is disabled, is NVLS still turned on?
>
> Yes, NVLS (SHARP) is still turned on. But based on our experiments, the results are almost the same for NCCL_NVLS_ENABLE=0 and NCCL_NVLS_ENABLE=0.
>
> > What is the precision used during the inference?
>
> We use bf16 precision for all the experiments.
>
> > What is the memory of a single H100 GPU? For Figure 2, when Batch Size = 64, why there is no number reported for TP=1 and 2
>
> For batch size 64, we run into OOM with the models in tp size = 1, 2 cause we were using cuda_graph + flash attention for benchmarking. When we generate static cuda graph for our 8B model, we observe high memory consumption and OOM as below:
> ```
> [rank0]:     static_next_token = prefill(model, static_x.view(batch_size, -1), static_input_pos, **sampling_kwargs)
> [rank0]:     return func(*args, **kwargs)
> [rank0]:     logits = model(x, input_pos)
> [rank0]:     return self._call_impl(*args, **kwargs)
> [rank0]: torch.OutOfMemoryError: CUDA out of memory. Tried to allocate 15.66 GiB. GPU 0 has a total capacity of 79.11 GiB of which 4.97 GiB is free. Including non-PyTorch memory, this process has 74.13 GiB memory in use. Of the allocated memory 41.05 GiB is allocated by PyTorch, with 13.52 GiB allocated in private pools (e.g., CUDA Graphs), and 32.02 GiB is reserved by PyTorch but unallocated. If reserved but unallocated memory is large try setting PYTORCH_CUDA_ALLOC_CONF=expandable_segments:True to avoid fragmentation.  See documentation for Memory Management
> ```
> With our new setup under torch.compile, we can run BS=64 with all the TP sizes, as expected from the reviewer’s calculation. Results can be found at https://anonymous.4open.science/r/ICLR2025_rebuttal-B81D/8B%20-%20With%20NVLink%20-%201024x512%20-%20torch_compile.PNG
>
> > For results in Table 3, it seems like Parallel Transformer is catching up with Ladder Transformer (Average from -0.17 to +0.12, Wikitext PPL from -0.53 to -0.06), as the number of parameters scales up from 1B to 3B, this raises concerns on the scalability of Ladder Transformer. Providing additional experiments with a larger model size or an explanation for why this trend might not continue could be helpful in supporting the significance of the Ladder Transformer.
>
> It is difficult to conclude the scaling trend from two data points (1B and 3B) and we agree with the reviewer that pretrain more models would make this clear. However, just the 3B pretraining run already costs 1228 H100 hours for us, and pretraining a larger model is beyond our computation budgets at this point.

---

> ### Author Response · Authors · 2024-11-21
> **Clarification 2**
>
> > In Figure 2, for the case Without NVLink, how to explain the reason of a throughput drop of batch size 4 and 16, when the TP world size shifts from 2 to 4?
>
> We reran the code and generated an updated version of Fig. 2(2) with a larger sequence length (512) to confirm the observed performance degradation, as shown in https://anonymous.4open.science/r/ICLR2025_rebuttal-B81D/8B%20-%20Without%20NVLink%20-%201024x512%20-%20no_compile.PNG. The performance degradation happens similarly as shown in the paper when batch_size = 16.
> We hypothesize the following reasons for the degradation when tp-size = 4:
> 1. Increased Communication Overhead: A larger TP size leads to more communication between GPUs, which becomes costly, especially without NVLink, as seen in the Fig. 2(2) results. This lack of NVLink results in higher latency and more expensive data transfer.
> 2. GPU Latency Bound: A larger TP size reduces memory I/O and compute load per GPU. Lower utilization prevents GPUs and optimized kernels from operating at their full potential.
> As a result, the performance benefit from reduced computation volume and latency with tp-size = 4 is outweighed by the increased communication overhead. This leads to diminished overall performance gains.
>
> > How to understand the correlation of the 3 columns in Table 2?
>
> We agree with your opinion on the relationship between prefill, decode, and end-to-end speedup and we thank the reviewer for pointing out the weird inconsistency in that entry. After verifying the results, we found that we swapped the data of prefill and decode somehow, the log file output as below (gpt-dense is the standard Transformer):
> For gpt-dense, we got: prefill: 15.89ms, decode: 172.39ms, token/sec: 169.53;
> For upper-bound we got: prefill: 10.01ms, decode: 122.94ms, token/sec: 240.16;
> Thus in the paper, the prefill improvement is 37.00(%) and the decode improvement is 28.69(%).
>
> To double-verify, we rerun NVLINK-UpperBound-Llama-8B with two settings (a. Cuda_Graph - Flash_Attention which is the same as the paper, b. Torch.compile), and found that both results agree with the formula the reviewer concludes. Again, we appreciate the detailed reading of the reviewer that helped us detect this subtle mistake.
>
> | | Prefill-Improvement | Decode-Improvement | Token/sec Improvement |
> | ----------- | ----------- | ----------- | ----------- |
> | CudaGraph + FlashAttention | 37.42% | 29.68% | 42.23 |
> | Torch compile | 37.69% | 40.96% | 69.16 |
>
> > It would be better to have a breakup of computation (both MLP and Attention) and communication (both with and without NVL) to help understand the importance of the overlapping, as well as how much overlapped can be, and is achieved with Ladder Transformer.
>
> We agree with the reviewer that providing detailed speedup can provide a better landscape on where the Ladder Residual gained its largest setup, and such results can potentially shape the architecture design. We will do more detailed analysis on this in the future.

---

> > ### Comment · Reviewer_Jbio · 2024-11-22
> > **Feedback for Rebuttal**
> >
> > Thank you for your response. Below is my feedback for each thread:
> >
> > ---
> >
> > > Benchmarking speedup under faster framework, and results for 70B and 405B
> >
> > - I assume that **bs=4,tp=1** is actually **bs=4,tp=4** because that pattern applies for all other batch sizes, and tp=1 does not make sense to any speedup (since you don't have communication in that case);
> > - I do not agree with the author's claim that when model grows larger and necessitates a multi-node hosting, using TP cross node would make the work more important. The author totally ignores the common practice of using PP for cross-node communication in this case;
> > - The latest absolute number (2463 tok/s/GPU) looks reasonable to me. However, with NVLink, the new table seems to suggest that Ladder Transformer only has a marginal improvement against Parallel attn:
> >   - For tp=4, the improvement against Parallel attn is at most 2.8% faster (bs=4, with compile);
> >   - For tp=8, it is 9.5% (bs=4, with compile);
> >   - With a larger batch size, the improvement is getting worse. For example, for the `tp=8, with compile` setup, from `bs=4` to `bs=64`, the improvement against Parallel Attn drops from 9.5% to 1.8%. For `tp=4, with compile`, when `bs=64`, it is already worse than Parallel Attn.
> >   - For a larger model size (70B), the same trend applies. For bs=16, tp=8, LadderTransformer is only 3.8% better than Parallel Attn.
> > - The file https://anonymous.4open.science/r/ICLR2025_rebuttal-B81D/README.md does not provide me any data. The raw format only has three paragraphs of pure text;
> > - My concern on using too many GPUs for a small model and small batch size is still unsolved;
> >
> > Answering the following questions can help improve the paper's significance:
> >
> > - Is there a range of setup including (model size, batch size, TP, ...), where LadderTransformer is significantly better than Parallel Attn? If so, how to prove the importance of that domain;
> > - Adding experiments for (8B, bs=128,256...) could help with my concern on the fact that computation is too sparse;
> > - Comparing the result of LadderTransformer with TP+PP when hosting a model on multiple nodes, instead of using a cross-node TP.
> >
> > > compatibility with PP
> >
> > I acknowledge the author's example code of integrating LadderTransformer with PP. However, my main concern is that the communication volume is three times larger than traditional transformer in this case: the `residual`, `current_mlp_out`, and `current_attention_out`.
> >
> > Having an quantitive explanation on the impact of such an communication could address this concern. For example, the author could try to prove that the extra communication overhead is much lower than that saved by using async AllReduce.
> >
> > > Clarification 1
> >
> > The response solves my concern about NVLS and precision.
> >
> > For OOM, such an error message does not give me extra information. However, since the author has figured out an alternative, this implementation detail is no longer a concern. (See the first part for my demand of experiment with a larger batch size)
> >
> > For the model performance not scalable, I'm still concerned on it, especially given the fact that LadderTransformer already shows the scalability issue on the throughput when comparing with Parallel Attn
> >
> > > Clarification 2
> >
> > The update of Table 2 looks reasonable. The explanation for the behavior when switching from tp=2 to tp=4 is still not convincing:
> > 1. For communication overhead, the default behavior of NCCL is to pipeline packets ([ref](https://images.nvidia.com/events/sc15/pdfs/NCCL-Woolley.pdf)) during the communication. In this way, the communication time does not change much when the number of GPU grows;
> > 2. For sparsity, a lower batch size should be even more sparse, while that unique phenomenon only applies for bs=16 but not larger.

---

> > > ### Author Response · Authors · 2024-11-25
> > > **Author Responses**
> > >
> > > > Comparing the result of LadderTransformer with TP+PP when hosting a model on multiple nodes, instead of using a cross-node TP.
> > >
> > > We are implementing PP for our method and benchmarking setup and we might not be able to finish this within the rebuttal period, but we will post results if we happen to get them. Note that PP is restricted by the batch size, otherwise we are simply pipelining the models to different nodes without speedup from using mini-batch. Therefore, our contribution of pure-TP speedup in cross-node setup still has its use case.
> > >
> > > > I acknowledge the author's example code of integrating LadderTransformer with PP. However, my main concern is that the communication volume is three times larger than traditional transformer in this case: the residual, current_mlp_out, and current_attention_out.
> > > Having an quantitive explanation on the impact of such an communication could address this concern. For example, the author could try to prove that the extra communication overhead is much lower than that saved by using async AllReduce.
> > >
> > > The pseudocode provided is not optimal enough. It should be noted that we can update the residual (residual = residual + current_attention_out)  before passing to the next pipeline stage thereby only requiring to pass 2 tensors: residual, current_mlp_out.
> > >
> > > It should also be noted that the tensor current_mlp_out is not needed until attention computation is finished. This can thus be sent asynchronously to the next pipeline stage and thus only the updated residual tensor is the tensor blocking the communication.
> > > Moreover, we see that when only 1 token with batch size 8 (decode phase) is being communicated across nodes (IB connected), the communication time is latency bound irrespective of the fact if 3 tensors of shape (8, 1, 16384) or 1 tensor of shape (8, 1, 16384) are transmitted. The communication time in both cases is ~2e-6 seconds and independent of message size. This is not the case during prefill but prefill constitutes a much smaller percentage of the generation time.

---

> > > ### Author Response · Authors · 2024-12-04
> > > **TP+PP results, large batch size result, new pareto-frontier plot**
> > >
> > > We collected some additional results over the past few days. The new results are added to https://anonymous.4open.science/r/ICLR2025_rebuttal-B81D/70B%20-%20With%20NVLink%20-%201024x32%20-%20no_compile.PNG as well. Hopefully they can clarify some of your concerns!
> > >
> > > > Is there a range of setup including (model size, batch size, TP, ...), where LadderTransformer is significantly better than Parallel Attn? If so, how to prove the importance of that domain;
> > >
> > > We re-drew the Pareto-frontier plot here - https://anonymous.4open.science/r/ICLR2025_rebuttal-B81D/8B%20-%20Pareto-frontier%20-%20torch_compile.png. We can see that the boundary of the ladder arch is faster than the parallel architecture in the latency requirement < 4ms. It's also expected that on devices with bad networking setup Ladder Residual will show a larger speedup. Here when we considered 16 GPUs scenario for serving 405B - https://anonymous.4open.science/r/ICLR2025_rebuttal-B81D/405B%20-%20PP2%20+%20TP8%20-%20torch_compile.png, we can also see that ladder is consistently better than the parallel architecture. Again, while we acknowledge that the margin over parallel is not huge, parallel itself isn't an adopted approach, and when used in PaLM, is not designed to speed up parallelism. We consider it as an alternative approach to reduce communication and showed that Ladder Residual is better in both performance and speed.
> > >
> > > > Adding experiments for (8B, bs=128,256...) could help with my concern on the fact that computation is too sparse;
> > >
> > > For 8B, at bs=128, the parallel baseline achieves 1.158x speedup while Ladder Residual archives 1.145x, confirming the trend that as batch size increases, the parallel eventually becomes better at speed. However, Ladder Residual still shines in a lot of setups and can be found in our results and our response above.
> > >
> > > > Comparing the result of LadderTransformer with TP+PP when hosting a model on multiple nodes, instead of using a cross-node TP.
> > >
> > > We provide the speedup of TP+PP at https://anonymous.4open.science/r/ICLR2025_rebuttal-B81D/405B%20-%20PP2%20+%20TP8%20-%20torch_compile.png. Pipeline parallelism can be easily applied on top of Ladder Residual and we show that Ladder Residual is outperforming the parallel baseline and is able to give around 11% speedup on 405B compare with the standard Transformer, which is significant.

---

> ### Author Response · Authors · 2024-11-25
> **Author Responses**
>
> We thank the reviewer's time for responding to our rebuttals. Here are our responses for further addressing the concerns:
>
> > The file https://anonymous.4open.science/r/ICLR2025_rebuttal-B81D/README.md does not provide me any data. The raw format only has three paragraphs of pure text;
>
> The anonymous link we linked to is an anonymous GitHub repo, you should be able to find other files in it. Either on the left-hand side or the top side there should be a list of files in the repo and those are our additional benchmarking results. Please let us know if you still cannot find it though!
>
> > I assume that bs=4,tp=1 is actually bs=4,tp=4 because that pattern applies for all other batch sizes, and tp=1 does not make sense to any speedup (since you don't have communication in that case);
>
> Yes bs=4, tp=4 is correct, thanks for catching that.
>
> > My concern on using too many GPUs for a small model and small batch size is still unsolved;
>
> Using too many GPUs for a small model can generally come into play when the small model (8B for example) is being used as a speculator for a larger model (say 405B). In this scenario, since the larger model is already running on a large number of GPUs, it makes sense to run the small model on a large number of GPUs as well. In the meantime, we also show speedup on larger models.
>
> > Is there a range of setup including (model size, batch size, TP, ...), where LadderTransformer is significantly better than Parallel Attn? If so, how to prove the importance of that domain;
>
> From https://anonymous.4open.science/r/ICLR2025_rebuttal-B81D/70B%20-%20With%20NVLink%20-%201024x512%20-%20torch_compile.PNG, we are consistently better than the parallel attn/mlp baseline on 70B model with TP=8 from bs=1 to bs=32. Indeed we observe a smaller gap as the batch size increases (9.05% more speedup than parallel attn/mlp with bs=1, to 4.81% with bs=32), this is a current limitation of our method, but in theory we shouldn’t observe this much degradation. We are planning to investigate this in the future.
>
> > Adding experiments for (8B, bs=128,256...) could help with my concern on the fact that computation is too sparse;
>
> We are running these experiments. From https://anonymous.4open.science/r/ICLR2025_rebuttal-B81D/8B%20-%20With%20NVLink%20-%201024x512%20-%20torch_compile.PNG we can also infer that at bs > 128, ladder-residual is likely to only have similar speedup as parallel attn/mlp, but we will provide results for completeness.
>
> We want to note that parallel attn/mlp as an alternative approach isn’t what the community has adopted, and in our experiment, we found that ladder-residual performs better in terms of model accuracy on 1B size and similar on 3B size compare with the parallel attn/mlp architecture. Ladder Residual shows significant speedup against the standard Transformer, and a modest gain over an alternative approach. It also has the potential to be directly converted from a pre-trained standard Transformer. We believe these results make Ladder Residual a better alternative worth being recognized.

---

### Official Review · Reviewer_GtCs · 2024-11-04

**Soundness:** 3
**Presentation:** 3
**Contribution:** 3
**Rating:** 6
**Confidence:** 3

**Summary:**

Typical implementations of large language model inference use tensor parallelism to distribute across multiple GPUs. However, tensor parallelism suffer from high communication overhead when deploying to multiple GPUs. This paper proposes a new transformer-based network architecture for large language models, which is able to reduce communication overhead in tensor parallelism without introducing significant accuracy drop. The experiments show that the proposed architecture can achieve 29% speedup on 8 GPUs.

**Strengths:**

* The proposed architecture can reduce communication overhead while incurring negligible overhead
* The proposed architecture is simple to implement and requires no low-level code modification

**Weaknesses:**

* The paper has no evaluation when scale up to more GPUs.
* The paper has no evaluation in conjunction with other existing parallelisms.

**Questions:**

Thanks for the submitting the excellent paper to ICLR. However, I have a few concerns about the evaluation section of the paper.

First, the paper only evaluates with TP size up to 8. However, the paper has no evaluation when scale up to more than 8 GPUs. It is thus obscure how the performance of the proposed technique when scaling up to multiple GPU nodes.

Second, the paper has no consideration how the proposed technique behave when using other parallelisms. While the proposed architecture should work orthogonally with other parallelisms, it would be much speedup the proposed method could provide when used in conjunction with other parallelisms.

---

> ### Author Response · Authors · 2024-11-21
> **Benchmarking speedup with TP=16, and discussion about other parallelisms**
>
> > First, the paper only evaluates with TP size up to 8. However, the paper has no evaluation when scale up to more than 8 GPUs. It is thus obscure how the performance of the proposed technique when scaling up to multiple GPU nodes.
>
> We only benchmarked TP size up to 8 in the paper since cross-node TP becomes expensive and we didn’t get time to evaluate that. Since our method hides the communication latency behind computation, it allows us to reconsider cross-node TP as a potential approach for a very large model.
>
> For 405B, even loading the model in bf16 requires > 800GB GPU memory, therefore > 8 GPU is needed. **Here we benchmarked llama-3.1-405B under TP=16 (2 nodes, each with 8 H100 GPUs) and obtain a larger speedup compare with 8B or 70B** as the saving from communication becomes a larger portion of the end-to-end latency. Our method still outperforms the parallel attn/mlp baseline across batch size while the model accuracy is higher on the smaller models we tested (training a 405B model is out of our computation budget).
>
> ---
>
> | | bs=1 | bs=4 | bs=8 | bs=16 |
> | ----------- | ----------- | ----------- | ----------- |  ----------- |
> | Ladder Residual | 1.364x | 1.308x | 1.393x | 1.349x |
> | Parallel attn/mlp | 1.238x | 1.242x | 1.286x | 1.272x |
>
> ---
>
> See https://anonymous.4open.science/r/ICLR2025_rebuttal-B81D/README.md for more details and compare with 8B and 70B results
>
>
> > Second, the paper has no consideration how the proposed technique behave when using other parallelisms. While the proposed architecture should work orthogonally with other parallelisms, it would be much speedup the proposed method could provide when used in conjunction with other parallelisms.
>
> One of the advantages of our proposed architecture is that it solely modifies the input/output of each module and therefore can be combined with other parallelism as using them on a standard transformer. Pipeline Parallelism needs a bit special treatment, which can be combined with our proposed model architecture by synchronizing before the pipeline boundary and sending the tensors to the next pipeline stage (please refer to our response to Reviewer Jbio for the detail). The expected speedup should be the same as if we apply them on a standard Transformer.

---

> > ### Comment · Reviewer_GtCs · 2024-11-26
> >
> > I would appreciate the authors for the response. I have read it and it answers all my questions. I would also suggest the author to include the above discussion in the paper to make it even stronger.

---

### Official Review · Reviewer_zw4s · 2024-11-04

**Soundness:** 3
**Presentation:** 3
**Contribution:** 2
**Rating:** 5
**Confidence:** 3

**Summary:**

This paper proposes Ladder Residual, which is a modified model architecture of Transformer models to enable better overlapping between computation and communication of the Tensor Parallelism execution.
It reroutes the output of Attention/MLP output to the block after the next block.
In this way, as for the Tensor Parallelism, one of the dependencies between the AllReduce and block computation is eliminated and thus can be overlapped to achieve better performance.
This paper trains the 1B and 3B models to show that it achieves good accuracy compared to the naive Transformer block structure.
It conducts the performance comparison of 1B, 3B and 8B models on up to 8 GPUs to demonstrate is performance efficacy.

**Strengths:**

- The rerouting of the residual to break the dependency of computation and communication is novel.
- It conducts the experiments of both accuracy and efficiency to show the effectiveness of both.

**Weaknesses:**

- It uses up to 8B model for the motivation illustration and evaluation. However, it is less likely to use 8 H100 GPUs to inference the small 8B model in practice.
- It could be better to have a discussion and comparison to the model comparison scheme, which can also serve the LLM efficiently with high accuracy.

**Questions:**

- Why not conduct the experiment on 70B models? I believe this can be a better model that worth the multi-GPU serving. But given the high compute-intensity of the 70B model, the ratio of the AllReduce overhead can be smaller than 8B, which can be a concern to the motivation of this paper.
- I suggest having some discussion and comparison to quantization. For example, the int4 weight quantization can make a 70B model running on a single H100 GPU, without losing too much accuracy.
- Having a comparison to the pipeline parallelism can also better demonstrate the contribution of this paper. Note some recent studies have shown good performance of pipeline parallelism than the pure TP (https://blog.vllm.ai/2024/07/23/llama31.html).

---

> ### Author Response · Authors · 2024-11-21
> **Benchmarking speedup on 70B and 405B, and analysis on the speedup trend as we scale model sizes.**
>
> > It uses up to 8B model for the motivation illustration and evaluation. However, it is less likely to use 8 H100 GPUs to inference the small 8B model in practice.
>
> We have benchmarked 70B size in table 1 under the setup of 1024 prompt length, 256 generated token, TP=8 and batch size 1, and observed 17% tokens/second improvement and 30% without NVLink.
>
> In the response below, we benchmarked 70B and 405B more thoroughly to demonstrate the effectiveness of Ladder Residual at larger model size.
>
>
>
> > Why not conduct the experiment on 70B models? I believe this can be a better model that worth the multi-GPU serving. But given the high compute-intensity of the 70B model, the ratio of the AllReduce overhead can be smaller than 8B, which can be a concern to the motivation of this paper.
>
> To provide a more comprehensive results, beyond the one setup in Table 1 that shows the effectiveness of our method on 70B model, we additionally benchmarked it under various batch sizes . As shown in https://anonymous.4open.science/r/ICLR2025_rebuttal-B81D/70B%20-%20With%20NVLink%20-%201024x32%20-%20no_compile.PNG, **With NVLink, the speedup on 70B is lower due to higher compute-intensity as the reviewer points out, but is still significant and our method outperforms the baseline.** For example, the improvement goes from *1.297x* in 8B to *1.188x* in the batch size 1 case with 32 decoded tokens.
>
>
>
> However, **if we increase decode tokens to 512 and disabled NVLink P2P communication, the communication volume is larger and sometimes outweighs the downward trend going from 8B to 70B due to increasing compute-intensity, leading to higher speedup.** Below for batch size 4, at TP=4 8B observes a larger speedup while at TP=8 70B has a larger speedup.
>
> ---
>
> | | tp=2 | tp=4 | tp=8 |
> | ----------- | ----------- | ----------- |  ----------- |
> | Ladder Residual 8B | 1.163x | 1.327x | 1.457x |
> | Ladder Residual 70B |  OOM | 1.259x | 1.610x |
>
> ---
>
> To enhance our experimental results, we also provide benchmarking results with 512 decoding tokens on 70B and 405B with various batch sizes and TP sizes as below.
> Here we benchmark with torch.compile as it is more memory efficient than our previous benchmarking setting (there is a recent feature that allows us to benchmark with compile). The compile setup brings higher speedup due to reduced non-communication overhead but the overall trend stays the same (please refer to our response to Reviewer Jbio for more detail).
>
> ---
>
>
> 70B results, NVLink=True:
> | | bs=1, tp=4 | bs=1, tp=8 | bs=4, tp=8 | bs=16, tp=8 |
> | ----------- | ----------- | ----------- | ----------- |  ----------- |
> | Ladder Residual | 1.111x | 1.244x | 1.240x | 1.188x |
> | Parallel attn/mlp | 1.057x | 1.120x | 1.152x | 1.144x |
>
> ---
>
> For 405B, even loading the model in bf16 requires > 800GB GPU memory, therefore we only benchmark under TP=16 (2 nodes, each with 8 H100 GPUs) for various batch sizes. Below are results with NVLink=True:
>
> | | bs=1 | bs=4 | bs=8 | bs=16 |
> | ----------- | ----------- | ----------- | ----------- |  ----------- |
> | Ladder Residual | 1.364x | 1.308x | 1.393x | 1.349x |
> | Parallel attn/mlp | 1.238x | 1.242x | 1.286x | 1.272x |
>
> See https://anonymous.4open.science/r/ICLR2025_rebuttal-B81D/README.md for results under more batch size, tp size, as well as results without NVLink.
>
> ---
>
> **Our method (Ladder Residual) still achieves significant speedup for both 70B and 405B and consistently outperforms the parallel attn-mlp baseline on both speed and model quality (shown on smaller size in the paper due to computation constraints)**. Note that when we use TP=16, where the communication needs to happen across nodes, the speedup is larger as communication is more expensive, and this leads to much larger speedup despite the increased compute-intensity from 70B to 405B. As we have seen that models are getting larger and larger, we expect that these techniques that decouple computation and communication will become more important.

---

> ### Author Response · Authors · 2024-11-21
> **Discussion about other model compression scheme**
>
> > It could be better to have a discussion and comparison to the model comparison scheme, which can also serve the LLM efficiently with high accuracy.
>
> Lots of post-training model compression techniques have been proposed for LLM, for example N:M sparsity, quantization, or depth-wise pruning. These techniques are orthogonal to our proposed methods and can be applied in combination. Our contribution is a modification of computation flow that makes tensor parallelism more efficient, and does not negatively interfere with other techniques. We leave for the future work to study the performance impact of applying compression techniques on our Ladder Residual architecture.
>
>
> > I suggest having some discussion and comparison to quantization. For example, the int4 weight quantization can make a 70B model running on a single H100 GPU, without losing too much accuracy.
>
> Given the ever increasing model capacity, quantization is a very viable option however, even with quantization, it’s impossible to run larger models like Llama 405B on a single H100 80GB GPU. Using tensor parallelism to run large models is still an approach the community actively uses and our contribution is to make it faster. Our work is orthogonal to quantization and it should be noted that our proposed model architecture can be easily combined with current SOTA quantization approaches.
>
>
> > Having a comparison to the pipeline parallelism can also better demonstrate the contribution of this paper. Note some recent studies have shown good performance of pipeline parallelism than the pure TP (https://blog.vllm.ai/2024/07/23/llama31.html).
>
> In the post, a combination of pipeline parallelism (PP) and tensor parallelism (TP) are used to run 405B models on 16 GPUs. **Our method makes the TP portion faster while still being compatible with PP**. Due to the expensive cross-node communication, TP is usually not used across nodes, but our methods provide an opportunity to reconsider the trade-off and potentially scale TP to multiple nodes. The best practice of multi-dimensional parallelism under different scenarios is still a resolved question for researchers and practitioners, and as long as TP is still being used, our method can provide efficiency gain.

---

> > ### Comment · Reviewer_zw4s · 2024-11-26
> >
> > Thanks for the clarification and the evaluation on the larger models. It definitely requires a lot of efforts. But I still have some concerns for the current shape of the paper.
> > - Quantization is technically orthogonal to the parallelism strategy. However, quantization can make many of the parallelism unnecessary. For example, the 4-bit (GPTQ, AWQ) or even 6-bit (Quant-LLM) can make the 70B model execute on a single A100/H100. 405B model indeed requires the model parallelism, but this paper has not demonstrate that the accuracy of the large model is still good with the architecture modification.
> > - I still expect the exact performance comparison between TP and PP. Given that PP can have less communication traffic, is it possible to only PP to achieve higher performance than TP proposed in this paper?

---

> > > ### Author Response · Authors · 2024-12-04
> > > **Author responses**
> > >
> > > > Quantization is technically orthogonal to the parallelism strategy. However, quantization can make many of the parallelism unnecessary. For example, the 4-bit (GPTQ, AWQ) or even 6-bit (Quant-LLM) can make the 70B model execute on a single A100/H100. 405B model indeed requires the model parallelism, but this paper has not demonstrate that the accuracy of the large model is still good with the architecture modification.
> > >
> > > While it is true that 405B can be served on a single node with quantization, there can eventually be cases that cross-node is necessary as the model size grows. We also want to note that the purpose of using Tensor Parallelism is not solely for saving memory, but also to increase speed. Tensor Parallelism is one of the most widely supported parallelism for inference and is independent of the batch size, which makes it applicable in every scenario. In this paper, we propose Ladder Residual which tackles the communication bottleneck in TP and makes it even faster. Quantization can be applied on top of the Ladder Residual and it is the user's choice to decide whether to use multiple GPUs or not. We acknowledge the effectiveness of quantization but we believe these are two parallel research directions and the progress in each of them is going to make an impact.
> > >
> > > > I still expect the exact performance comparison between TP and PP. Given that PP can have less communication traffic, is it possible to only PP to achieve higher performance than TP proposed in this paper?
> > >
> > > In both https://anonymous.4open.science/r/ICLR2025_rebuttal-B81D/70B%20-%20PP%20+%20TP%20comparison%20-%20torch_compile.png and https://anonymous.4open.science/r/ICLR2025_rebuttal-B81D/405B%20-%20PP2%20+%20TP8%20-%20torch_compile.png, we show that when using PP+TP, Ladder still gives significant speedup and outperforms the parallel attn/mlp. PP is more often used in cross-node comparison, and can only bring benefit to global throughput instead of single batch's decode & prefill latency. As there are lots of optimizations that can be done for PP, we don't make a direct comparison in our preliminary implementation. We do believe TP is a widely adopted choice in the community and is more flexible than PP due to its independence of batch size. In the benchmarking discussion the reviewer provides https://blog.vllm.ai/2024/07/23/llama31.html, TP=8 is also being used but combined with PP=2 as cross-node communication is slower.

---

### Official Review · Reviewer_93XY · 2024-11-04

**Soundness:** 2
**Presentation:** 3
**Contribution:** 3
**Rating:** 6
**Confidence:** 3

**Summary:**

This paper introduces Ladder Residual and Desync Residual, two architectural modifications to the original transformer architecture. The authors aim to address the communication latency bottleneck inherent in TP by decoupling computation from communication, enabling overlapping operations. The Ladder Residual method is tested on both scratch-trained and adapted models, demonstrating competitive performance compared to traditional architectures. Additionally, the concept of Desynced Residual is introduced to further mitigate communication overhead in low-connectivity settings.

**Strengths:**

* Given communication is usually the bottleneck in adopting Tensor Parallelism (especially in settings with low bandwidth interconnects), the proposed methods can significantly reduce the amount of communication required and mitigate the problem.
* The evaluations are comprehensive, conducted across various settings and benchmarks.
* The paper is well-written and easy to follow.

**Weaknesses:**

* Implementing Ladder Residual may require substantial retraining or adaptation efforts for existing models, which may be challenging for larger models;
* The evaluation sections are restricted to models with relatively small sizes (up to 8B).

**Questions:**

Regarding the Evaluation section:
- why are the results consistently better without nvlink than with nvlink, across all three baselines? I would assume Megatron-style TP would perform better with NVlink.
- In Fig 2(2), why is there a performance degradation when TP world size =4?
- Is the baseline "standard transformer" using data parallelism or Megatron-style tensor parallelism?

Regarding the Hybrid-Ladder technique:
- What are the considerations when deciding to apply ladder-residual on the upper half (or later half) of the transformer layers?

---

> ### Author Response · Authors · 2024-11-21
> **Benchmarking speedup on 70B and 405B**
>
> > The evaluation sections are restricted to models with relatively small sizes (up to 8B).
>
> For efficiency, we did benchmark 70B size in table 1 under one setup and we agree with the reviewer that tensor parallelism is more interesting to study on larger models. To provide more comprehensive results, we benchmarked 70B, 405B with various batch sizes and TP sizes below, all with NVLink:
>
> ---
>
> 70B speedup results:
> | | bs=1, tp=4 | bs=1, tp=8 | bs=4, tp=8 | bs=16, tp=8 |
> | ----------- | ----------- | ----------- | ----------- |  ----------- |
> | Ladder Residual | 1.111x | 1.244x | 1.240x | 1.188x |
> | Parallel attn/mlp | 1.057x | 1.120x | 1.152x | 1.144x |
>
> ---
>
> For 405B, even loading the model in bf16 requires > 800GB GPU memory, therefore we only benchmark under TP=16 (2 nodes, each with 8 H100 GPUs) for various batch sizes.
>
> | | bs=1 | bs=4 | bs=8 | bs=16 |
> | ----------- | ----------- | ----------- | ----------- |  ----------- |
> | Ladder Residual | 1.364x | 1.308x | 1.393x | 1.349x |
> | Parallel attn/mlp | 1.238x | 1.242x | 1.286x | 1.272x |
>
> See https://anonymous.4open.science/r/ICLR2025_rebuttal-B81D/README.md for results under more batch size, tp size, as well as results without NVLink.
>
> ---
>
> **Our method (Ladder Residual) still achieves significant speedup for both 70B and 405B and consistently outperforms the parallel attn-mlp baseline on both speed and model quality (shown on smaller size in the paper due to computation constraints)**. Note that the relative improvement of 70B is smaller than 8B size, since the computation scales faster than communication as model size increases (bits to be communicated scale linearly, while computation can scale quadratically). However, when we use TP=16, where the communication needs to happen across nodes, the improvement is larger as communication is more expensive. As we have seen that models are getting larger and larger, we expect that these techniques that decouple computation and communication will become more important.
>
> We are currently running the experiment of adapting llama-3.1-70B-Instruct to Ladder Residual and will update the results if we are able to finish it within the rebuttal period.

---

> ### Author Response · Authors · 2024-11-21
> **Response on adaptation strategy and retraining cost**
>
> > Implementing Ladder Residual may require substantial retraining or adaptation efforts for existing models, which may be challenging for larger models;
>
> With only 2 epochs of supervised finetuning on 1.6 billion tokens (1-2 days on an H100 node with an off-the-shelf finetuning library), we can recover the performance of llama-3.1-8b-instruct while gaining 14.5% speedup, which makes it more efficient serve the model for months. This is significantly cheaper than the trillion-tokens scale pre-training cost. (we could not find the report on post-training token count for Llama-3, but it is likely more costly than our setup, and we don’t require human preference data or RL training). Given that inference is starting to account for a significant fraction of compute, we think this is a favorable tradeoff. More importantly, the new architecture offers much more flexibility in terms of hardware to serve the model as we have decoupled the computation and communication (NVLink). We are excited to see how this changes the networking required for LLM inference.
>
> > What are the considerations when deciding to apply ladder-residual on the upper half (or later half) of the transformer layers?
>
> We found that it’s difficult to apply ladder-residual on all the layers for a pre-trained LLM, and used the zero-shot result to decide which layers to apply. Our hypothesis is that a lot of knowledge is stored within the lower half of the layer, and with a light-weight retraining (1.6B token in our case), we can’t expect the fine-tuning dataset to contain the lost knowledge. As the field as a whole develops more understanding on how knowledge is stored in these models, we expect it would be easier to apply these architectural changes to pretrained models.

---

> ### Author Response · Authors · 2024-11-21
> **Clarifications**
>
> > why are the results consistently better without nvlink than with nvlink, across all three baselines? I would assume Megatron-style TP would perform better with NVlink.
>
> We are reporting the relative speedup instead of the absolute speed in the paper. It’s correct that without NVLink, we would observe worse absolute speed. Since without NVLink the communication is slower, our technique that overlaps communication with computation is expected to lead to a larger relative improvement than with NVLink as we reported.
>
>
> > In Fig 2(2), why is there a performance degradation when TP world size =4?
>
> We reran the code and generated an updated version of Fig. 2(2) with a larger sequence length (512) to confirm the observed performance degradation, as shown in https://anonymous.4open.science/r/ICLR2025_rebuttal-B81D/8B%20-%20Without%20NVLink%20-%201024x512%20-%20no_compile.PNG. The performance degradation happens similarly as shown in the paper when batch_size = 16.
> We hypothesize the following reasons for the degradation when tp-size = 4:
> 1. Increased Communication Overhead: A larger TP size leads to more communication between GPUs, which becomes costly, especially without NVLink, as seen in the Fig. 2(2) results. This lack of NVLink results in higher latency and more expensive data transfer.
> 2. GPU Latency Bound: A larger TP size reduces memory I/O and compute load per GPU. Lower utilization prevents GPUs and optimized kernels from operating at their full potential.
> As a result, the performance benefit from reduced computation volume and latency with tp-size = 4 is outweighed by the increased communication overhead. This leads to diminished overall performance gains.
>
> > Is the baseline "standard transformer" using data parallelism or Megatron-style tensor parallelism?
>
> The baseline transformer uses standard Megatron-style Tensor Parallelism.

---

### Author Response · Authors · 2024-12-04
**Summary of discussion points during the rebuttal period and re-emphasize the focus of our paper**

Dear reviewers,

We have incorporated some (due to space limit) of the new results into the revised version. The full results on various batch sizes and TP sizes can be found at https://anonymous.4open.science/r/ICLR2025_rebuttal-B81D/README.md. To address reviewers’ concerns on how much speedup our Ladder Residual can have on larger models, we provided the benchmarking results on 70B and 405B models, showing that Ladder Residual outperforms the parallel attn/mlp architecture along with a large speedup over the standard transformer (eg: 24% on 70B batch size 64 with NVLink, around 30% on 405B batch size 16 with NVLink). The benchmarking results on 405B especially demonstrated the advantage of our method on cross-node Tensor Parallelism (TP) which can be necessary for large model 405B.

While pipeline parallelism (PP) is not a primary focus of our study and can be combined with the Ladder Residual TP seamlessly, we provide the benchmarking results on 70B (single-node serving) and 405B (cross-node serving) to show that: as ladder can be applied on top of TP + PP, it also leads to speedup in all settings when we combine two parallelisms, and is able to outperform the parallel baseline. We also include the results of PP + TP in https://anonymous.4open.science/r/ICLR2025_rebuttal-B81D/README.md. Ladder Residual is able to achieve 10-15% improvement over the standard Transformer on 70B size, across batch sizes up to 64, and around 1.15x speedup for the 405B model when we restrict TP to be intra-node.

Finally, we want to re-emphasize that the goal of our paper is an architecture-level modification that allows overlapping the GPU communication and computation. Despite a lot of previous efforts on optimizing the communication, **to our knowledge we are the first paper that proposes to change the model architecture to create overlapping opportunities, without touching low-level kernels, making it easily deployable on any hardware.** We showed that such architecture modification is performing on-par with the standard transformer. As model size grows, multi-gpu or even cross-node serving will become more and more important, and our paper provides a fresh perspective on designing the architecture with communication optimization in mind. Such design can be applied to any architecture that is inherently sequential, although in this paper we only conducted experiments on Transformer-based language models due to its popularity.

Our approach is orthogonal to other efficient language model methods or parallelism and can be seamlessly combined while still enjoying the benefit of more efficient Tensor Parallelism.

We thank all the reviewers for the valuable feedback and suggestions, incorporating them made our paper much more clear.

---

### Meta-Review · Area_Chair_SVUY · 2024-12-09

**Metareview:**

The paper proposes an architectural modification to Transformer models, called Ladder Transformer, that improves the performance of tensor parallel training.

While the idea is interesting, the empirical validation was not sufficiently convincing. The initial version of the paper did not provide large-scale experiments to verify that the Ladder Transformer architecture meets or exceeds the accuracy of standard Transformers. Although the author rebuttal provided throughput and speedup numbers for 80B and larger models, accuracy results - which are the most crucial for any work in which a novel model architecture is proposed - were not provided.

It is currently unclear whether the Ladder Transformer architecture is amenable to pipeline parallelism, even though the authors provided new results on larger models. Carefully-designed ablation studies would be required to convince readers that there are no unexpected costs to combining pipeline parallelism with Ladder Transformer.

Ultimately, the scope of the paper's results might demonstrate that Ladder Transformer is effective at the 7B model scale, but it is unclear whether it remains effective at 80B or above (especially in terms of accuracy). If the paper was written with a complete set of large-model experiments (including accuracy results), that would be a step in the right direction.

**Additional Comments On Reviewer Discussion:**

The initial version of the paper did not provide large-scale experiments to verify that the Ladder Transformer architecture meets or exceeds the accuracy of standard Transformers. The author rebuttal provided throughput and speedup numbers for 80B, but not accuracy results, which are the most important.

In the pipeline parallel discussion between reviewer Jbio and the authors, there were new experiments claiming that Ladder Transformer can be successfully integrated with pipeline parallelism. However, reviewer Jbio pointed out that the pipeline parallel communication cost of Ladder Transformer may be unfavorable compared to standard well-optimized Transformer code. This issue would need to be addressed by more careful ablation studies.

---

### Decision · Program_Chairs · 2025-01-22

Reject